# Irritable Bowel Syndrome-Like Disorders in Endometriosis: Prevalence of Nickel Sensitivity and Effects of a Low-Nickel Diet. An Open-Label Pilot Study

**DOI:** 10.3390/nu12020341

**Published:** 2020-01-28

**Authors:** Raffaele Borghini, Maria Grazia Porpora, Rossella Casale, Mariacatia Marino, Emilia Palmieri, Nicoletta Greco, Giuseppe Donato, Antonio Picarelli

**Affiliations:** 1Department of Translational and Precision Medicine, Sapienza University, 155 00161 Rome, Italy; raffaele.borghini@uniroma1.it (R.B.); rosscas@hotmail.it (R.C.); mariacatia.marino@uniroma1.it (M.M.); nicoletta.greco25@gmail.com (N.G.); 2Department of Gynecology, Obstetrics and Urology, Sapienza University, 155 00161 Rome, Italy; mariagrazia.porpora@uniroma1.it (M.G.P.); emilia.palmieri93@gmail.com (E.P.); giuseppe.donato@uniroma1.it (G.D.)

**Keywords:** nickel allergy, allergic contact mucositis, endometriosis, irritable bowel syndrome (IBS), low-nickel diet

## Abstract

Alimentary nickel (Ni) may result in allergic contact mucositis (ACM), whose prevalence is >30% and may present with IBS-like and extra-intestinal symptoms. These symptoms are also frequent in endometriosis, and Ni allergic contact dermatitis has already been observed in endometriosis. Therefore, intestinal and extra-intestinal symptoms in endometriosis may depend on a Ni ACM, and a low-Ni diet could improve symptoms. We studied the prevalence of Ni ACM in endometriosis and focused on the effects of a low-Ni diet on gastrointestinal, extra-intestinal, and gynecological symptoms. We recruited 84 women with endometriosis, symptomatic for gastrointestinal disorders. Thirty-one out of 84 patients completed the study. They underwent Ni oral mucosa patch test (omPT), questionnaire for intestinal/extra-intestinal/gynecological symptoms, and a low-Ni diet. Clinical evaluation was performed at baseline (T0) and after three months (T1). Twenty-eight out 31 (90.3%) patients showed Ni omPT positive results, with Ni ACM diagnosis, whereas three out of 31 (9.7%) patients showed negative Ni omPT. After three months of low-Ni diet, all gastrointestinal, extra-intestinal and gynecological symptoms showed a statistically significant reduction. Ni ACM has a high prevalence in endometriosis and a low-Ni diet may be recommended in this condition to reduce gastrointestinal, extra-intestinal and gynecological symptoms.

## 1. Introduction

### 1.1. Nickel and Allergic Contact Mucositis 

Nickel (Ni) is a ubiquitous element in nature. It is involved in many human physiological processes, but at high concentrations, it may also be toxic, inducing inflammatory and allergic disorders in susceptible individuals. Ni can be ingested, can come into contact with the skin or be inhaled, and Ni blood-level concentration reflects its degree of exposure [1].

The main source of Ni for humans is food: Ni levels in food range from < 0.1 to 0.5 mg/kg and depend on Ni content in the soil and water, industrial production and packaging process [2]. Ni-rich foods include tomato, cocoa, licorice, beans, mushrooms, broad-leafed vegetables, whole-wheat flour, soy, corn, onion, garlic, shellfish, nuts, canned food, and tea [3]. Moreover, other foods such as maize, are rich in Ni because of fertilizers [4].

Exposure to Ni can lead to multiple clinical manifestations. General awareness of Ni allergy is progressively increasing, as well as some environmental risk factors, such as industrialization of food and modern dietary patterns, thus Ni allergy now represents a major health and socio-economic issue: Its prevalence can be even ≥ 30% [5].

Allergic Contact Dermatitis (ACD) is the best-known Ni-related clinical manifestation, and its diagnosis is based on Ni epicutaneous patch test (ePT). Apart from dermatological manifestations, Ni-related symptoms in Ni-sensitive patients are mainly gastrointestinal, such as abdominal pain, swelling, diarrhea, constipation, oral ulcers, nausea, vomiting and gastro-esophageal reflux disease (GERD). More recently, Ni Allergic Contact Mucositis (ACM) has been suggested as a possible cause of irritable bowel syndrome (IBS)-like symptoms: It is a low-grade gastrointestinal inflammation due to alimentary exposure to Ni and its diagnosis is based on Ni oral mucosa patch test (omPT). Both Ni ACD and ACM can be included in the so-called “systemic Ni allergy syndrome” (SNAS) [5].

What is more, extra-intestinal manifestations of “Ni sensitivity” may include respiratory (rhinitis, asthma), neurological (headache), uro-genital (interstitial cystitis and vulvo-vaginitis) and general disorders (osteo-articular pain, asthenia, chronic fatigue syndrome, fibromyalgia) [6,7,8]. On this purpose, a low-Ni diet has already been demonstrated useful in reducing Ni-related symptoms [5,9]. 

### 1.2. Endometriosis

Endometriosis is a chronic, estrogen-dependent inflammatory disorder characterized by the presence and growth of endometrial tissue outside the uterine cavity [10]. Common locations of endometriosis are the pelvic peritoneum, the ovaries, the recto-vaginal septum, and also the abdominal cavity, including the gastro-intestinal tract [11]. The estimated prevalence of endometriosis is 2%–10% in the general population of women of childbearing age, but it can even rise up to 30%–33% in infertile women or women with chronic pelvic pain [12].

Endometriosis is a chronic multifactorial disease, influenced by genetic, hormonal, immunological, and environmental factors. Risk factors are family history, long menstrual cycle, low parity, and poor physical activity [13]. Environmental pollution seems to increase its prevalence [14,15]: Specific substances of natural origin (phytoestrogens) or environmental toxicants (xenobiotics) are able to affect sex hormones and the immune system. Specifically, some heavy metals can interfere with estrogenic activity, resulting in a new class of xenoestrogens defined as “metalloestrogens”, such as Ni [16]. In this regard, Ni can bind estrogen receptors (ERs) and induce the proliferation of ERα+ breast cancer cells [17,18]. Moreover, higher Ni concentrations have been found in serum and ectopic endometrial tissue from patients with endometriosis [19]. In addition, women with endometriosis already showed a slightly higher rate of Ni allergic contact dermatitis vs. control patients [20]. 

The main symptoms of endometriosis are dysmenorrhea, chronic pelvic pain, dyspareunia, irregular uterine bleeding, and infertility. It can be also a disabling condition, with chronic fatigue, depression, and gastrointestinal disorders [21]. Symptoms are often not related to stage and location of endometriotic lesions [22], and their causes are still unclear: Inflammation, tissue infiltration, hyperalgesia, hormonal and/or neuropathic factors are under investigation [23,24,25]. Gastrointestinal disorders have also been identified as possible causes or exacerbating factors, probably involving spasms of levator ani or other pelvic floor muscles [26,27].

Endometriosis diagnosis is based on history, symptoms and signs, diagnostic imaging, and laparoscopy, which allows biopsy sampling and histological confirmation. The combination of laparoscopy and histological examination is still considered the gold standard for the diagnosis of endometriosis [28]. However, the role of the surgery has been recently discussed [29].

Unfortunately, to date, no single therapy is available as a sure successful option and recommendation for endometriosis. Treatment may be surgical, but symptoms can recur [30,31]. Progestins, oral contraceptives, and gonadotropin-releasing hormone (GnRH) agonists are largely used and they can be effective on symptoms and can reduce recurrences [32]. However, disease recurrence is observed in more than 50% of cases after treatment discontinuation [33]. Recently, several new experimental drugs have been proposed (e.g., selective estrogen receptor modulators, anti-estrogens, aromatase inhibitors), but they are not suitable for a long-term treatment [34].

### 1.3. Gastrointestinal Disorders and Endometriosis

Gastrointestinal symptoms such as abdominal pain, swelling, constipation, and diarrhea are frequent in endometriosis and even tend to get worse during the menstrual period [35]. There is still no agreement about the relationship between gastrointestinal symptoms and endometriosis. In the case of “bowel endometriosis”, in which endometrial cells infiltrate the bowel, these symptoms are common and can be attributed to local prostaglandin-mediated inflammation, mechanical intestinal obstruction and/or recurrent microhemorrhages [36,37,38,39]. Gastrointestinal symptoms in endometriosis may also be due to an alteration of the enteric nervous system, which is responsible for controlling the muscular and secretory activity of the intestinal tract, the reproductive system and the urinary tract [40]. In this complex neural system, any gastrointestinal inflammatory stimulus in the pelvic cavity could affect the functioning and responses of other organs and vice versa [41,42]. 

Moreover, there seems to be a relationship between intestinal pathophysiology and the endocrine system, in particular, sex hormones. In fact, several studies have shown the presence of GnRH-, LH- and FSH receptor and immunoreactive neurons in the intestinal tract and other pelvic organs: This could explain changes in intestinal symptoms during the menstrual cycle [43,44].

Gastrointestinal symptoms in endometriosis may also be related to IBS or IBS-like disorders [45]. Recent studies showed a higher prevalence of Ni skin allergy in women with endometriosis, as well as a possible Ni involvement in its etiopathogenesis [20,46,47]. On these bases, it is possible to hypothesize that an IBS-like disorder, such as Ni ACM, may be the cause or contributing factor of gastrointestinal symptoms in women with endometriosis.

On these premises, our aims were: 1) To study the prevalence of Ni ACM in women suffering from endometriosis and gastrointestinal symptoms, 2) to evaluate the effects of a low-Ni diet on gastrointestinal and extra-intestinal symptoms in women with endometriosis.

## 2. Materials and Methods

### 2.1. Patients

Study design: Pilot study. Eighty-three patients referred to the Department of Gynecology and Urology were consecutively recruited between October 2016 and February 2018. Inclusion criteria were: Fertile age, diagnosis of endometriosis by diagnostic imaging and/or laparoscopy, presence of at least three gastrointestinal symptoms with score ≥ 5 on the modified GSRS questionnaire completed at T0, in order to exclude less significant clinical pictures. Exclusion criteria were: History of past or current cancer, inflammatory bowel disease, celiac disease, IgE-mediated food allergies, infectious diseases. 

The study was performed in compliance with the Declaration of Helsinki. Approval of the local ethics committee was obtained (study approval: Report n.54/2019 of the Board of the Department of Internal Medicine and Medical Specialties, Sapienza University of Rome). Written informed consent was obtained from all patients.

### 2.2. Nickel Oral Mucosa Patch Test

Once included in the study (T0), all patients underwent Ni omPT to detect the presence of Ni ACM. Ni omPT is a useful tool for Ni ACM diagnosis and consists of a 5-mm filter paper disk saturated with a 5% solution of Ni sulfate in Vaseline (0.4 mg Ni-sulfate/8 mg Vaseline). It is applied on the upper lip mucosa and held in place by a transparent adhesive film (Tegaderm, 3M), while a paper disk with only 8 mg Vaseline is also applied as a control test. After two hours of exposure, Ni omPT can induce type IV hypersensitivity reactions: Specific local alterations on labial mucosa of Ni-sensitive patients can be visible, such as edema and hyperemia, probably related to a TLR4-dependent innate immune response. Aphthous/vesicular lesions may also appear even after 24–48 hours as late reactions. Late general symptoms triggered by omPT (such as swelling and abdominal pain, diarrhea, headache, foggy mind, itching, dermography) should also be considered [47]. 

### 2.3. Symptom Questionnaire

The Gastrointestinal Symptom Rating Scale (GSRS) questionnaire modified according to “Salerno’s experts’ criteria” is standardized and currently used in the diagnostic protocol for non-celiac gluten sensitivity. It can provide an objective evaluation of gastrointestinal and extra-intestinal symptoms, both during free diet and after restrictive diet [48].

The questionnaire consists of a list of gastrointestinal symptoms (abdominal pain, heartburn, acid regurgitation, bloating, nausea, borborygmus, swelling, belching, flatulence, decreased or increased evacuations, loose or hard stools, urgent need for defecation, oral/tongue ulcers) and extra-intestinal symptoms (dermatitis, headache, foggy mind, fatigue, numbness of the limbs, joint/muscle pain, fainting). For our purposes, typical symptoms of endometriosis (dysmenorrhea, dyspareunia, and pelvic pain) have also been added to the questionnaire. They all are associated with a numeric scale (score ranging from 0 to 10) that represents the perceived intensity. The questionnaire has been administered at enrollment time (T0) and after three months of low-Ni diet (T1).

### 2.4. Low-Nickel Diet

After Ni omPT and the first symptom questionnaire, all patients followed a balanced low-Ni diet for three months. Since Ni is present in almost all foods and its absolute removal from the diet is impossible, we recommended to avoid only foods with an estimated high content of Ni (Table 1) [49]. 

The quantity of Ni in food is extremely variable as it is conditioned by many factors: Quantity of Ni in soil and irrigation water, the plant species, any fertilizers and/or pesticides used. 

Moreover, patients were asked to avoid the use of stainless-steel utensils to reduce Ni contamination during cooking. Patients wrote a daily dietary diary and compliance with a balanced elimination diet was evaluated by trained dieticians (biweekly telephone interviews) [48]. 

### 2.5. Statistical Analysis

Data obtained during the present study were both qualitative (omPT results) and quantitative (modified GSRS questionnaire). Qualitative data were expressed as frequencies (both absolute and relative). Quantitative data were first analyzed by means of D’Agostino–Pearson omnibus test to verify the normal distribution within each statistical sample. Since some of the resulting p-values were significant (p <0.05), data obtained from patients did not reasonably belong to Gaussian distributions, and therefore, the results were expressed as mean ± standard deviation (SD) and processed by non-parametric tests. Comparisons between the scores of each symptom detected before and after treatment were performed using the Wilcoxon signed-rank test for paired data. P values <0.05 were considered significant.

Statistical analysis has been performed using GraphPad’s Package Prism version 5.2 (GraphPad Software Inc., San Diego, CA).

The flow chart of the study design is summarized in Figure 1a.

Study management, Ni omPT, administration of the modified GSRS questionnaire, patient follow-up, and final data processing have been performed at the Department of Internal Medicine and Medical Specialties/Department of Translational and Precision Medicine. 

## 3. Results

### 3.1. Patients

Of the 83 patients who completed the questionnaire at T0, 51 were recruited as they met the criterion of at least three gastrointestinal symptoms with a score ≥ 5. Four out of the total 51 patients have been excluded: Three were affected by celiac disease and one was affected by wheat allergy. Sixteen out of the remaining 47 patients dropped out of the study. These patients mostly stated that they had difficulty following dietary restrictions, although the proposed observation time (three months) was not excessively long. The main reasons for dropping out were: Personal inability to renounce or reduce certain foods of common consumption (e.g., cocoa, tomato, corn, broad-leaved vegetables, dried fruit), difficulty or unwillingness to look for food alternatives, restrictions, and changes affecting the whole family food environment. Therefore, a total of 31 patients (age range 19-46, mean age 33.5 years) completed the study (Figure 1b). The clinical features of the patients recruited are summarized in Table 2.

### 3.2. Nickel Oral Mucosa Patch Test

Twenty-eight out of the 31 patients studied (90.3%) showed omPT positive results and received Ni ACM diagnosis, while three out of 31 (9.7%) showed omPT negative results (Table 2).

Patients with positive Ni omPT showed marked local mucosal lesions (erythema, edema, and/or vesicles) within two hours after patch application (Figure 2a,b). Moreover, gastrointestinal and extra-intestinal systemic symptoms have been recorded 2-48 hours after the test.

Patients with negative Ni omPT results showed no local or systemic symptoms (Figure 2c,d).

### 3.3. Symptom Questionnaire

All 15 gastrointestinal symptoms showed a statistically significant decrease in intensity (p <0.05) after three months of low-Ni diet (T1) (Figure 3). 

All seven extra-intestinal symptoms showed a statistically significant reduction at T1 (*p* < 0.05). Finally, all gynecological symptoms analyzed showed a significant improvement at T1 compared to T0 (*p* < 0.005) (Figure 4).

## 4. Discussion

Ni is a ubiquitous metal in nature and, once ingested, it can be responsible for a specific ACM, a low-grade inflammatory gastrointestinal disorder with also extra-intestinal implications. Its prevalence is estimated to be over 30% [5]. Symptoms triggered by the ingestion of Ni-rich foods in Ni-sensitive patients are mainly gastrointestinal, but they can also be systemic, involving the skin, the nervous and the reproductive system [49]. A low-Ni diet can improve symptoms in Ni-sensitive patients, though foods high in Ni are largely consumed, especially in the Mediterranean diet, such as tomatoes, cocoa, beans, mushrooms, broadleaf vegetables, and whole flour [47]. As Ni ePT is useful for the diagnosis of Ni ACD, so Ni omPT has proved to be a specific and sensitive tool for Ni ACM diagnosis [5,47,49,50,51].

Endometriosis is a chronic disease characterized by dysmenorrhea, chronic pelvic pain, and dyspareunia. The etiopathogenesis of chronic pain and other symptoms in endometriosis is still unclear: The poor correlation with lesions’ localization and extent may suggest that hormonal, neurologic and systemic inflammatory factors may be involved [13]. Since environmental pollutant exposure is considered a possible cause of endometriosis, exposure to Ni has been recently suggested as an additional risk factor [14,52]. Moreover, heavy metals seem to interfere with estrogenic activity, thus identifying a new class of xenoestrogens defined as “metalloestrogens” and Ni falls into this category [53]. However, Ni allergy could also be only an important cause of abdominal pain in women with endometriosis without having a causal effect on the disease.

Women with endometriosis often complain of IBS-like symptoms, but there is still an open debate about their relationship [54]. Intestinal localization of endometriosis lesions increased visceral hypersensitivity, alterations of the myenteric plexus or the hormonal regulation may be considered [55]. Concerning IBS-like disorders, recent studies showed a higher rate of Ni allergy in endometriosis than in the control population [20]. On the other hand, since Ni omPT revealed to be more sensitive and specific than ePT in detecting adverse reactions to Ni-rich foods, the prevalence of Ni sensitivity in endometriosis may be even higher [5].

Our study is pioneering in this regard since it has been conducted on women with endometriosis and gastrointestinal symptoms by means of Ni omPT. Our data showed a Ni ACM prevalence of about 90.3% in this specific category of patients, but it could have been resulted even higher: The already mentioned operator-dependent limits of Ni omPT would have been overcome by the use of Laser Doppler Perfusion Imaging (LPDI), as already demonstrated in our previous work [47].

Another goal of our study was to focus on the effects of a low-Ni diet in endometriosis and our results were more than encouraging: After the administration of a low-Ni diet for a period of only three months, we obtained a significant improvement of all intestinal and extra-intestinal symptoms, including those typical of endometriosis (chronic pelvic pain, dysmenorrhea, and dyspareunia). Our findings show for the first time that Ni ACM may have a causative role in this clinical picture. 

On the other hand, our study is limited by its observational nature and the lack of real randomized trial design. Moreover, the study comes from a single center and has been conducted on small sample size. This last aspect can be justified by the nature of this open-label pilot study and by the fact that many turned out to be dropouts. The causes of so many dropouts were mostly related to the incapacity or unwillingness to change eating habits. Furthermore, it has often been problematic for the patient to find family support in shopping, cooking, or storing food with a low Ni content. 

Another critical point may be the lack of accurate biomarkers and the impossibility to accurately measure Ni contained in food before and during the study: As previously mentioned, there is great variability of Ni content in foods, and there is still no technology capable of measuring it routinely. Thus, the only way to overcome this limit was to give all patients a balanced low-Ni diet on the base of an estimated average content of Ni in foods and under direct control of trained dieticians. A daily dietary diary and detailed interview also helped in monitoring adherence to the diet itself. 

According to recent literature, a low-FODMAP diet can also have possible beneficial effects on IBS-like symptoms (even in women with endometriosis) and, since a low-FODMAP diet can overlap with a low-Ni diet, this may lead to misinterpretation of our results [56]. However, at an even more careful analysis, the low-FODMAP diet includes not only a low-Ni diet, but also a low-lactose and a low-gluten diet, thus covering a large spectrum of high-prevalence pathologies, such as lactose intolerance and non-celiac gluten sensitivity: It is therefore highly predictable to obtain clinical benefits from a low-FODMAP diet, even if at the cost of probably not necessary dietary exclusions not supported by reliable diagnostic tests. In contrast, in our study, the suspicion of Ni-sensitivity was clinically relevant, as well as supported by a proper diagnostic test (Ni omPT) and a targeted diet therapy.

## 5. Conclusions

In conclusion, although our results should be interpreted with caution, our study supports the association between Ni-rich diet, Ni ACM, and symptomatic endometriosis. Considering the resulting high prevalence of Ni ACM in endometriosis and the relief from symptoms after a low-Ni diet, Ni ACM should be considered as the cause or contributing factor of intestinal and extra-intestinal symptoms in women with endometriosis. Moreover, a diet poor in Ni should be followed by this category of patients. If confirmed by further studies with larger populations and new possible validated biomarkers, our results could potentially change the clinical management of these high prevalence disorders.

## Figures and Tables

**Figure 1 nutrients-12-00341-f001:**
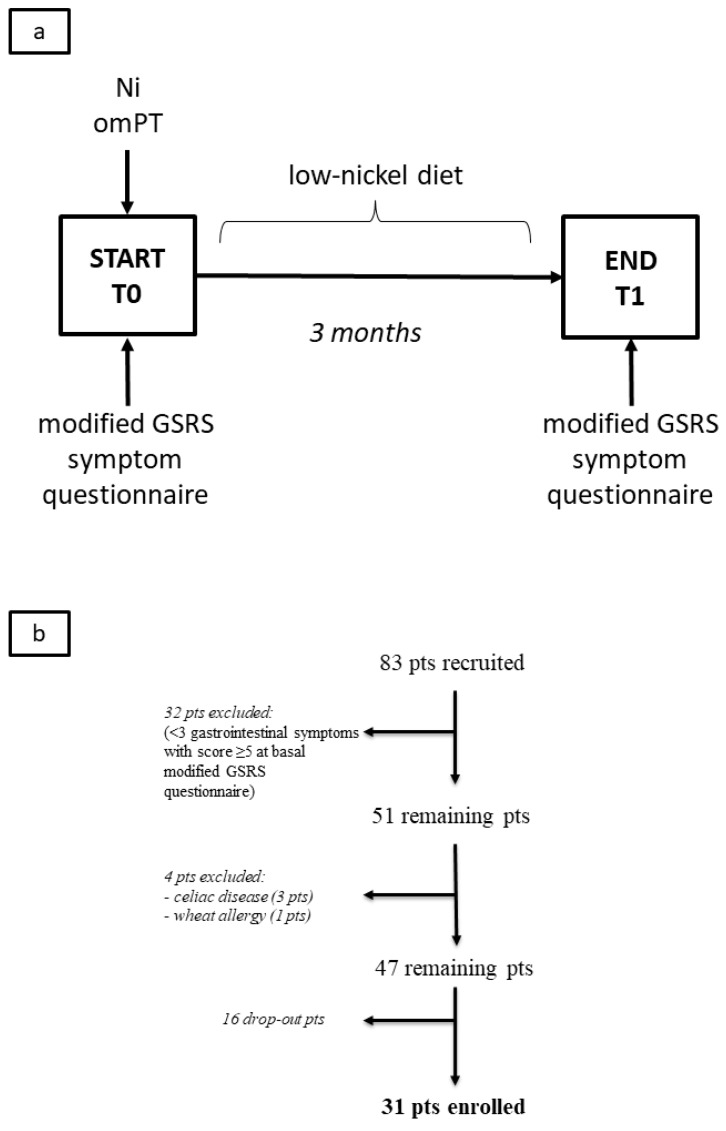
Flow charts of the study: (**a**) Study arrangement, (**b**) patient enrollment. Legend: *Ni omPT*, nickel oral mucosa patch test; *pts*, patients; *T0*, baseline; *T1*, after three months.

**Figure 2 nutrients-12-00341-f002:**
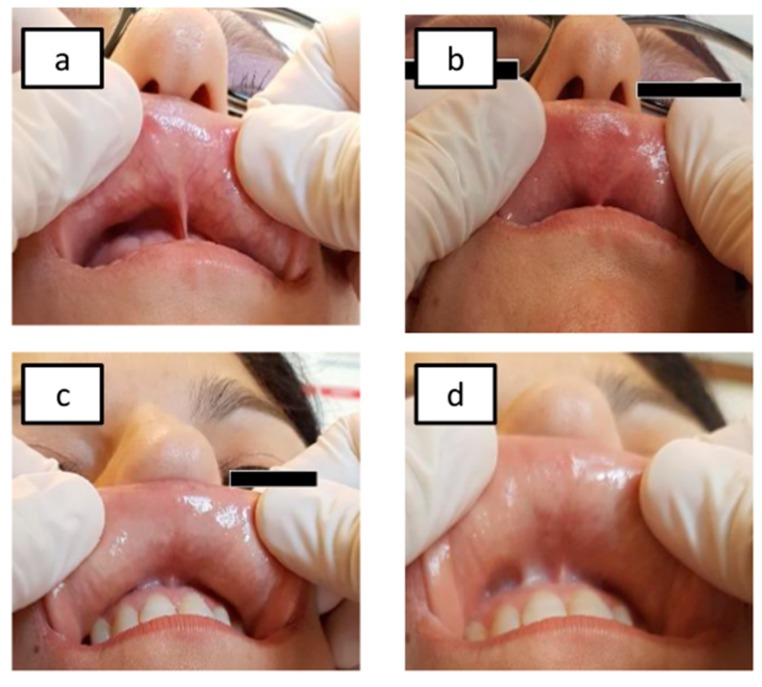
Ni omPT results: Ni-sensitive pts before Ni omPT application (**a**) and after Ni omPT removal (two hours) (**b**), non-Ni-sensitive pts before Ni omPT application (**c**) and after Ni omPT removal (two hours) (**d**). Legend: Ni omPT, nickel oral mucosa patch test; pts, patients.

**Figure 3 nutrients-12-00341-f003:**
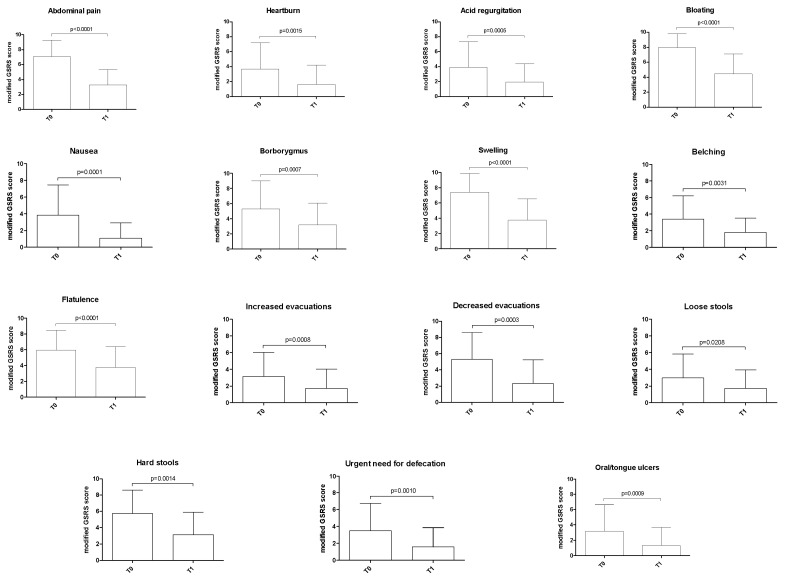
Variation of gastrointestinal symptoms after three months of low-nickel diet in women with endometriosis. The p-value was calculated using the Wilcoxon signed-rank test. Legend: *GSRS*, Gastrointestinal Symptom Rating Scale; *T0*, baseline; *T1*, after three months of low-nickel diet.

**Figure 4 nutrients-12-00341-f004:**
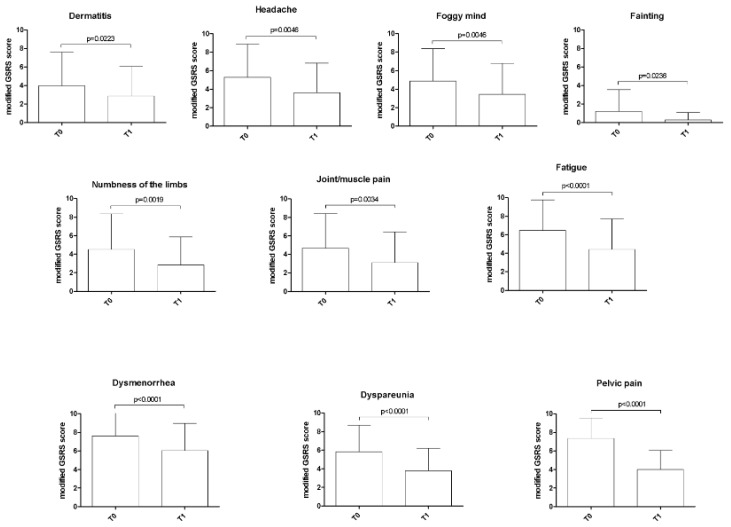
Variation of extra-intestinal and gynecological symptoms after three months of low-nickel diet in women with endometriosis. The p-value was calculated using the Wilcoxon signed-rank test. Legend: *GSRS*, Gastrointestinal Symptom Rating Scale; *T0*, baseline; *T1*, after three months of low-nickel diet.

**Table 1 nutrients-12-00341-t001:** Foods that contain a high amount of nickel [49].

Foodstuffs	Ni-containing foods
Fishes	Herring, mackerel, salmon, shellfish, tuna
Vegetables	Fresh and dried legumes (chickpeas, lentils, peanuts, peas, red kidney beans, soya beans, and soy products), garlic, green leafy vegetables (spinach), onion, raw carrots, tomatoes
Fruits	Fresh and dried fruits (almonds, hazelnuts, walnuts)
Cereals	Buckwheat, maize, millet, oat, rye, whole grain, whole wheat
Beverages	Beer, coffee, initial water flow from the tap (especially in the morning), red wine, tea
Others	Baking powder, canned foods, cocoa and chocolate, foods cooked in stainless steel utensils (especially if acidic foods as tomatoes), gelatin, linseeds, marzipan, Ni-containing vitamin supplements, strong licorice, sunflower seeds

Legend: *Ni*, Nickel.

**Table 2 nutrients-12-00341-t002:** Clinical features of the endometriosis patients at the time of recruitment and their Ni omPT results.

Pt n.	Age (yrs)	Endometriosis Duration (yrs)	Endometriotic Lesion’s Site	Size Score	Endometriosis Stage	Hormonal Treatment Duration	Surgery	Comorbidities	Concomitant Therapies	Ni omPT Result
1	38	3	LO EC	2	n.a.	-	-	-	-	Positive
2	25	2	rectum	n.a.	n.a.	EP (≥ 5 yrs)	-	PCOS	-	Positive
3	45	20	LO EC; POD; peritoneum	0	IV	-	LPS	folate-deficiency anemia	folic acid	Positive
4	31	5	LO EC; peritoneum	1	III	EP (≥ 5 yrs)	LPS	-	-	Positive
5	38	4	cesarean section scar	0	iatrogenic	EP (1-3 yrs)	LPT	generalized anxiety disorder	escitalopram	Negative
6	46	17	bilateral ECs; peritoneum	1	IV	EP (1-3 yrs)	LPS	-	-	Positive
7	31	4	bilateral ECs	1	IV	P (1-3 yrs)	LPS	-	-	Positive
8	49	21	bilateral ECs	2	IV	-	LPT; LPS	-	-	Positive
9	30	3	LO EC	2	n.a.	P (1-3 yrs)	-	-	-	Negative
10	32	3	LO EC	1	n.a.	EP (1-3 yrs)	-	-	-	Positive
11	39	20	RO EC	1	III	-	LPS	bilateral breast fibroadenomas	-	Positive
12	31	7	LO EC	0	n.a.	-	-	-	-	Positive
13	37	21	uterine cervix	0	n.a.	-	-	fibromyalgia	-	Positive
14	25	4	LO EC	2	III	P (1-3 yrs)	LPS	Hashimoto thyroiditis	-	Positive
15	27	3	RO; peritoneum	2	III	EP (1-3 yrs)	LPS	-	-	Positive
16	31	9	LO EC; peritoneum	2	III	EP (≥ 5 yrs)	LPS	-	-	Positive
17	38	3	peritoneum	0	III	EP (≥ 5 yrs)	LPS	-	-	Positive
18	25	4	LO EC	0	n.a.	-	-	-	-	Positive
19	19	3	peritoneum	0	n.a.	EP (1-3 yrs)	-	pollen allergy	-	Positive
20	43	4	bilateral ovarian ECs	1	n.a.	EP (< 1 yrs)	-	-	-	Positive
21	27	n.a.	LO EC	0	n.a.	-	-	-	-	Positive
22	40	20	bilateral ovarian ECs	0	n.a.	P (1-3 yrs)	-	PRL-secreting pituitary adenomas	cabergoline	Positive
23	35	10	RO EC	1	II	P (1-3 yrs)	LPS	hypothyroidism	levothyroxine	Negative
24	43	18	bilateral ovarian ECs	1	IV	P (1-3 yrs)	LPS	Hashimoto thyroiditis	-	Positive
25	27	1	RO EC	0	n.a.	-	-	psoriatic arthritis, fibromyalgia	-	Positive
26	20	1	3 LO ECs	1	III	EP (< 1 yr)	LPS	-	-	Positive
27	28	8	LO EC	0	III	P (≥ 5 yrs)	LPS	mild depressive disorder	amitriptyline	Positive
28	36	21	LO EC	0	n.a.	-	-	-		Positive
29	40	13	RO EC	0	IV	P (≥ 5 yrs)	LPS	Hashimoto’s thyroiditis	levothyroxine	Positive
30	38	2	bilateral ovarian ECs	2	III	EP (1-3 yrs)	LPS	fibromyalgia		Positive
31	24	1	3 RO ECs	0	n.a.	-	-	-		Positive

Legend: EP, estroprogestinic treatment; LO EC, left ovarian endometriotic cyst; LPS, laparoscopic surgery; LPT, laparotomy; n.a., not available; Ni omPT, Nickel oral mucosa Patch Test; P, progestogen; PCOS, Polycystic Ovary Syndrome; POD, Pouch of Douglas; PRL, Prolactin; Pt, patient; RO EC, right ovarian endometriotic cyst; yrs, years. Size score 0: < 30 mm; 1: 30-50 mm; 2: > 50 mm.

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
