# Peer review of "Irritable Bowel Syndrome-Like Disorders in Endometriosis: Prevalence of Nickel Sensitivity and Effects of a Low-Nickel Diet. An Open-Label Pilot Study"

_nutrients, 2020, doi:10.3390/nu12020341_

Round 1

Reviewer 1 Report

This is an interesting manuscript and the conducting study and obtained results are promising and could have a potential clinical implication. So on, the study is valuable what makes it worth publishing.

Some suggestions for authors:

Affiliations section: Wheather after Maria Grazia Popra should be „and”?

Line 43: „Ni can lead to multiple clinical manifestations.”– Ni or contact/exposure to Ni?

Line 44: „The prevalence of Ni allergy is in progressive growth” – Did authors considered whether the prevalence of Ni allergy is growing? Maybe not allergy but diagnosis of Ni allergy?

Line 66: „menstrual length cycle” shorter or longer?

Line 82-85: To my mind, the information about/description of diagnosis are not required. Authors should more concentrate on the lack of effective treatment – this problem explains the aim of the study.

Line 123: Why patients with cancer were excluded from the study? All type of cancer or hormone-dependent? There is some evidence linkage cancer with Ni allergy?

Line 125: According to guidelines of Nutrients,  authors should indicate the code of KE approval:

„ At a minimum, a statement including the project identification code, date of approval, and name of the ethics committee or institutional review board should be cited in the Methods Section of the article.”

As authors show in their manuscript, the NI- contact allergy is probably related to TL4/ response type of 4. What is the opinion about the nomenclature of reaction to Ni? Based on mechanism, it is rather Ni-hypersensitivity or food-specific reactivity?

Line 127: Whether the patients have been informed about the results of the examination of an abnormal reaction to Ni? It is important information, in regard to methodology and objective assessment severity of symptoms after the intervention.

Line 139 – GSRS scale – why authors did not refer to scale in the References section? Did the authors have permission to use the scale?

Line 142 – what is mean „free diet” – unrestricted eating?

Line 152: As the authors mentioned - Ni is present in almost all foods – how they assessed Ni exposure after following the diet? F.e. intake of Ni would be also dependent on daily calories intake – more calories – more foods with low Ni amount and the cumulative amount of the metal. Whether the authors estimated the amount of Ni intake before and after intervention based on dietary diaries?

Table 1: What is the source of information about main Ni sources – the presented table was create based on some book/an article? Please, including the source in the References section.

Table 2: Clinical features of the examined group should be rather presented as a median or frequency (%) and be more brief/compact. To my mind, the information of the table should including age/sex/duration of illness/intake of medication/comorbidities, at least.

Line 209: Why authors assess all gastrointestinal symptoms individually and do not compare the overall score of GSRS scale before and after intervention?

Results

The results section is too short.

Is it possible (based on dietary diaries) to assess intake of Ni and shows median/range of consumption before and after intervention? Compare these results with the severity of symptoms? Did the authors analyze the results after excluding patients without Ni allergy?

Author Response

REVIEWER 1

Comments and Suggestions for Authors

This is an interesting manuscript and the conducting study and obtained results are promising and could have a potential clinical implication. So on, the study is valuable what makes it worth publishing.

Some suggestions for authors:

Affiliations section: Wheather after Maria Grazia Porpora should be „and”?

Sorry, it was a typo and it has been removed.

Line 43: „Ni can lead to multiple clinical manifestations.”– Ni or contact/exposure to Ni?

As you suggest, “exposure to Ni” is more proper: proper substitution has been done.

Line 44: „The prevalence of Ni allergy is in progressive growth” – Did authors considered whether the prevalence of Ni allergy is growing? Maybe not allergy but diagnosis of Ni allergy?

Thank you for your comment. More probably, a greater awareness in the community around Ni allergy is the most important factor, as you suggest. On the other hand, we can’t also exclude a combination of some increasing environmental risk factors, such as industrialization of food and modern dietary patterns. Proper changes in the manuscript have been done.

Line 66: „menstrual length cycle” shorter or longer?

It is intended as "longer menstrual cycles". This clarification has also been added to the text.

Line 82-85: To my mind, the information about/description of diagnosis are not required. Authors should more concentrate on the lack of effective treatment – this problem explains the aim of the study.

As suggested, we more intensively stressed this lack (“Unfortunately, to date no single therapy is available as sure successful option and recommendation for endometriosis”).

Line 123: Why patients with cancer were excluded from the study? All type of cancer or hormone-dependent? There is some evidence linkage cancer with Ni allergy?

Patients with cancer have been excluded firstly for ethical reasons. Moreover, their cancer-related clinical conditions could have significantly affected the study results, resulting in a bias. On the other hand, this does not exclude the interesting suggestion (already present in the manuscript) regarding a possible relationship between Ni and hormones, as well as between Ni and hormone-dependent cancer: it could be the base for future very interesting work.

Line 125: According to guidelines of Nutrients, authors should indicate the code of KE approval:

„ At a minimum, a statement including the project identification code, date of approval, and name of the ethics committee or institutional review board should be cited in the Methods Section of the article.”

As requested, proper data has been added in the paper.

As authors show in their manuscript, the NI- contact allergy is probably related to TL4/ response type of 4. What is the opinion about the nomenclature of reaction to Ni? Based on mechanism, it is rather Ni-hypersensitivity or food-specific reactivity?

As showed in the paper, the best definition can be “Nickel Allergic Contact Mucositis” (Ni ACM), to be meant as “adverse reaction to dietary nickel” or, more precisely, “dietary Nickel-induced type IV hypersensitivity reaction”.

Line 127: Whether the patients have been informed about the results of the examination of an abnormal reaction to Ni? It is important information, in regard to methodology and objective assessment severity of symptoms after the intervention.

Yes, they have been properly informed about possible examination results (it was assessed in informed consent form, properly signed, as mentioned).

Line 139 – GSRS scale – why authors did not refer to scale in the References section? Did the authors have permission to use the scale?

As already declared in materials and methods section, The Gastrointestinal Symptom Rating Scale (GSRS) questionnaire is modified according to “Salerno's experts’ criteria” and is referred to reference n. 48.

Line 142 – what is mean „free diet” – unrestricted eating?

Yes, it is to be meant as “unrestricted diet”, in fact it is in opposition with “restricted diet” written immediately after.

Line 152: As the authors mentioned - Ni is present in almost all foods – how they assessed Ni exposure after following the diet? F.e. intake of Ni would be also dependent on daily calories intake – more calories – more foods with low Ni amount and the cumulative amount of the metal. Whether the authors estimated the amount of Ni intake before and after intervention based on dietary diaries?

As already stated in the paper, the only means currently available to follow this condition are avoiding/reducing foods with an estimated high content of Ni, a daily dietary diary and frequent interviews with trained dietitians to evaluate a proper compliance to a balanced low-Ni diet, without affecting a proper calorie intake (nickel in foods and calories are independent factors). These are the most effective tools currently available in common clinical practice. On the other hand, their objective limits are widely described in the discussion section (lines 278-284).

Table 1: What is the source of information about main Ni sources – the presented table was create based on some book/an article? Please, including the source in the References section.

As requested, proper reference source has been reported (ref. 49).

Table 2: Clinical features of the examined group should be rather presented as a median or frequency (%) and be more brief/compact. To my mind, the information of the table should including age/sex/duration of illness/intake of medication/comorbidities, at least.

As requested, table 2 has been made more compact and brief, trying to include the information requested.

Median or frequency data could be added, but in part such data is present in the text and/or would not add significant results for our purposes. Moreover, this would require an increase in the size of the table or even the addition of a new table.

Line 209: Why authors assess all gastrointestinal symptoms individually and do not compare the overall score of GSRS scale before and after intervention?

We considered it more appropriate and more scientifically correct to analyze each single symptom, in order to enhance any significant responses, possible trends or even possible worsening, even of a single symptom: reporting a cumulative GSRS score could have disguised interesting and significant results regarding isolated symptoms. Furthermore, the uniform and positive response of all the symptoms that we showed in the end was not obvious.

Results

The results section is too short.

The results section has been configured in this way to not result redundant, also considering related images with descriptions.

Is it possible (based on dietary diaries) to assess intake of Ni and shows median/range of consumption before and after intervention? Compare these results with the severity of symptoms?

Unfortunately, as already described within the limits of the work, it is not possible to obtain and report a precise quantity of nickel ingested or eliminated (it is the same problem that arises with the low-FODMAPs diet: see Halmos EP.When the low FODMAP diet does not work . J Gastroenterol Hepatol. 2017): there is no standardized method for precise monitoring. Although it was not possible to precisely correlate dietary Ni content and severity of Ni-related symptoms, dietary diaries allowed a proper monitoring of adherence to the low-nickel diet.

Did the authors analyze the results after excluding patients without Ni allergy?

As easily imaginable, during enrollment we could not predict such a high resulting prevalence (90.3 %) of NI-sensitive patients and, since only 3 patients finally presented Ni omPT negative results, we preferred to not exclude them and make a general evaluation of the results, giving due attention to the underlying endometriosis and the initial presence of IBS-like disorders in which Ni ACM can possibly occur.

Reviewer 2 Report

The authors selected patients with a diagnosis of endometriosis and at least 3 GI symptoms score at least 5  on the GRS scale. All were tested for mucosal responsiveness to Ni using a paper disk soaked with 5% NiSO4 applied to the upper lip mucosa. All patients followed a balanced low-Ni  diet for 3 months avoiding fish, a range of vegetables including legumes and beans along with dried fruit, wheat and other grains, beer, coffee and chocolate. However as the introduction makes clear Ni content depends on fertilisers used and can be much higher in some vegetable so that the diet is an imperfect way of reducing Ni intake. 90% of patients were positive by the Ni omPT which is very high compared to the literature. All GRS symptoms showed a decline over 3 months but without a placebo this could all be placebo effect.

General comment

This is a thought provoking study  and certainly of great interest but perhaps needs more cautious approach as this is a very preliminary finding

Title: This should reflect the uncontrolled preliminary nature of the data . Perhaps “an open label study of low Ni diet in endometriosis: a pilot study”

Introduction

More balance is needed in reviewing the literature. The RCT in ref 5 quoted uses ill defined endpoints asking patients to rate their “clinical condition” which makes it unclear what exactly is improving. I would like to see a more cautious approach emphasising the difficulty in doing trials in this area and the need to establish validated biomarkers and to do more better quality trials using objective endpoints. The estimated prevalence of endometriosis is 2-10%, but it can even rise up to 35-50% [12]. I find this confusing You  need to explain such disparities. There must be some reason why the prevalence can be so different. Some comment about the methods of ascertainment is needed which presumably explains the different estimates. Line 74 You state “In addition,  women with endometriosis already showed a high rate of Ni allergic contact dermatitis [20].  This is a bit misleading given the  odds ratio = 1.175; 95% confidence interval, 1.011-1.366. This is only just significant using a sample of around 10,000 suggesting the effect is weak. The quoted number of those with nickel allergy was 0.8%  v 0.3% in controls. I would not call this a high rate.  It would be better to give the figures.

Methods

Can you define what the criteria for positive test were? Did subjects have to develop all the manifestations listed or just one? Can you also give reproducibility data for this test and any validation data. Does it predict response to a low Ni diet? Are there any studies using the Salerno consensus measure that give repeatability or evidence of responsiveness or validity?  It seems to include some very poorly defined components such as “foggy mind “  One is left wondering how reproducible it is? All GRS symptoms showed a decline over 3 months but without a placebo this could all be placebo effect. What would be more persuasive would be evidence that the decline in those that were Ni-omPT positive v those that were negative. I would like to see this data in the text but recognise the study is underpowered to detect this which  would require recruiting more Ni-omPT negative patients. The prevalence of 90% Ni-omPT that you report is remarkably higher than that reported by Yuk. How can you explain this and are there any other  studies to compare with? Discussion You state that Ni-ACM prevalence is estimated to be over 30% but cite a review. I would like to see specific study cited as to me this seems much too high given that contact dermatitis only occurs in around 5% of the population and only a subgroup of those have the more severe syndrome. You need to be more circumspect as regards the evidence that a low Ni diet can improve symptoms since I cannot find a single randomised placebo controlled trials. Some statement about selection of patients is important. Patients with allergy are attracted / referred to doctors with special interests in allergy which tends to increase the apparent prevalence. The idea that Ni acts as a metalloestrogen is interesting but can you link Ni concentration in tissues to estrogenic effects?

Author Response

REVIEWER 2

Comments and Suggestions for Authors

The authors selected patients with a diagnosis of endometriosis and at least 3 GI symptoms score at least 5 on the GRS scale. All were tested for mucosal responsiveness to Ni using a paper disk soaked with 5% NiSO4 applied to the upper lip mucosa. All patients followed a balanced low-Ni diet for 3 months avoiding fish, a range of vegetables including legumes and beans along with dried fruit, wheat and other grains, beer, coffee and chocolate. However, as the introduction makes clear Ni content depends on fertilisers used and can be much higher in some vegetable so that the diet is an imperfect way of reducing Ni intake.

Indeed, the use of fertilizers can increase the nickel content of only specific vegetables and make dietary control more difficult. On the other hand, avoiding large and frequent consumption of those “risky” products seems to provide comforting data in clinical practice, in previous studies and also in this study.

90% of patients were positive by the Ni omPT which is very high compared to the literature.

This surprising result, although possibly influenced by the limits already described, can indeed significant in those selected patients with endometriosis and specific intestinal and extra-intestinal symptoms: thise prevalence may be causally related to endometriosis and/or to the complaints reported by patients with endometriosis.

Moreover, considerable importance is covered by the negative control test with vaseline performed in each patient.

All GSRS symptoms showed a decline over 3 months but without a placebo this could all be placebo effect.

From a practical, nutritional and ethical point of view, it was not easy to identify an alternative or placebo to the low-nickel diet. It may have been possible to think of a low-FODMAPs diet, but the relative limits and comparisons are already reported in the discussion section. In addition, the statistical significance obtained for all the 25 symptoms analyzed, may suggests a lower probability of placebo effect.

General comment

This is a tought provoking study and certainly of great interest but perhaps needs more cautious approach as this is a very preliminary finding.

Title: This should reflect the uncontrolled preliminary nature of the data . Perhaps “an open label study of low Ni diet in endometriosis: a pilot study”

Thank you for your suggestion. The title has been modified as follows: “Irritable Bowel Syndrome-like disorders in endometriosis: prevalence of nickel sensitivity and effects of a low-nickel diet. An opel-label pilot study”.

Introduction

More balance is needed in reviewing the literature. The RCT in ref 5 quoted uses ill defined endpoints asking patients to rate their “clinical condition” which makes it unclear what exactly is improving.

I would like to see a more cautious approach emphasising the difficulty in doing trials in this area and the need to establish validated biomarkers and to do more better quality trials using objective endpoints.

These observations are correct and already present in discussion section, as limits. As requested, we furtherly stressed them in discussion section. Our study even tries to go beyond reporting simple clinical data (albeit statistically significant), opening suggestive multidisciplinary scenarios that are still unknown.

The estimated prevalence of endometriosis is 2-10%, but it can even rise up to 35-50% [12]. I find this confusing You need to explain such disparities. There must be some reason why the prevalence can be so different. Some comment about the methods of ascertainment is needed which presumably explains the different estimates.

The percentage is 2-10% is in the general population of women of childbearing age, while the prevalence is greater in infertile women or in women with pelvic pain. Proper corrections and specifications have been added in the paper, as requested.

Line 74 You state “In addition, women with endometriosis already showed a high rate of Ni allergic contact dermatitis [20]. This is a bit misleading given the odds ratio = 1.175; 95% confidence interval, 1.011-1.366. This is only just significant using a sample of around 10,000 suggesting the effect is weak. The quoted number of those with nickel allergy was 0.8% v 0.3% in controls. I would not call this a high rate. It would be better to give the figures.

As requested, the sentence has been reformulated in a more balanced way.

Methods

Can you define what the criteria for positive test were? Did subjects have to develop all the manifestations listed or just one? Can you also give reproducibility data for this test and any validation data. Does it predict response to a low Ni diet?

All the manifestations listed in the text (combined, but also alone) are possible Ni omPT positive results and it is evident that they go deeper in the evaluation than the common skin patch tests: both local (especially erythema, edema and oral aphthous lesions) and systemic symptoms may occur and can be recorded.

The already discussed limit of Ni omPT lies in its operator-dependence, although in our previous works we have showed its diagnostic reliability and its clinical usefulness, along with clinical benefits deriving from a low- Ni diet in Ni sensitive subjects (please find reference 5, 47, 49).

In addition, the use of objective and reproducible laser doppler perfusion imaging (LDPI) has recently been combined with Ni omPT to support the reliability of Ni omPT itself, obtaining excellent results. In discussion section it is even mentioned the possibility that omPT may underestimate a positivity that can instead be read with LDPI.

Are there any studies using the Salerno consensus measure that give repeatability or evidence of responsiveness or validity?

Despite possible intrinsic limitations, the Salerno experts’ criteria represent today a sort of “gold standard” and an accurate tool for clinical assessment and monitoring of IBS-like disorders, such as non-celiac gluten sensitivity or nickel sensitivity. Furthermore, it is mainly based on the GSRS scale, an already standardized instrument with unquestionably objective and repeatable properties, which has been used in our work.

Catassi C, et al. Diagnosis of Non-Celiac Gluten Sensitivity (NCGS): The Salerno Experts' Criteria. Nutrients. 2015 Jun 18;7(6):4966-77. Casella G, et al. Non celiac gluten sensitivity and diagnostic challenges. Gastroenterol Hepatol Bed Bench. 2018 Summer;11(3):197-202. Review. Skodje GI, et al. Wheat challenge in self-reported gluten sensitivity: a comparison of scoring methods. Scand J Gastroenterol. 2017 Feb;52(2):185-192.

It seems to include some very poorly defined components such as “foggy mind”. One is left wondering how reproducible it is?

Although - as you say - it doesn't seem to be easily describable and reproducible, “foggy mind” can actually be considered one of the most reported systemic manifestations in IBS-like disorders, together with headache, fatigue, joint and muscle pain, leg or arm numbness, depression and many others.

Catassi C. Gluten Sensitivity. Ann Nutr Metab. 2015;67 Suppl 2:16-26.

All GRS symptoms showed a decline over 3 months but without a placebo this could all be placebo effect.

As mentioned above, a placebo effect is possible, although this theory contrasts with the surprising positive and significant results we have obtained regarding all the numerous symptoms analyzed.

What would be more persuasive would be evidence that the decline in those that were Ni-omPT positive v those that were negative. I would like to see this data in the text but recognise the study is underpowered to detect this which would require recruiting more Ni-omPT negative patients. The prevalence of 90% Ni-omPT that you report is remarkably higher than that reported by Yuk. How can you explain this and are there any other studies to compare with?

Discussion You state that Ni-ACM prevalence is estimated to be over 30% but cite a review. I would like to see specific study cited as to me this seems much too high given that contact dermatitis only occurs in around 5% of the population and only a subgroup of those have the more severe syndrome.

As already reported in the paper, SNAS has two possible manifestations: allergic contact dermatitis (ACD) and allergic contact mucositis (ACM). The first, more studied and known, has a prevalence that can even exceed 30% in Italy.

Schnuch A, et al. National rates and regional differ-ences in sensitization to allergens of the standard series. Population-adjusted frequencies of sensitization (PAFS) in 40,000 patients from a multicenter study (IVDK). Contact Dermatitis. 1997; 37 (5): 200–209. Torres F, et al. Management of contact dermatitis due to nickel allergy: an update. Clin Cosmet Investig Dermatol. 2009 Apr 17;2:39-48.

On the other hand, Ni ACM, which is less known and studied to date, has a much higher prevalence than the ACD in clinical experience: for this reason, the prevalence of our review reported in bibliography is prudently estimated ">30%".

It is not surprising that the prevalence of Ni ACM obtained in our study is >90% in selected patients with endometriosis and specific intestinal and extra-intestinal symptoms (see also previous comments on the same topic).

You need to be more circumspect as regards the evidence that a low Ni diet can improve symptoms since I cannot find a single randomised placebo controlled trials.

Unfortunately, there are no other works in the literature about Ni ACM, except those already mentioned in the bibliography, especially randomized placebo controlled trials: it is a completely new and extremely interesting topic, which probably deserves to be deepened and spreaded, given the encouraging preliminary results.

The fact that our work is not a randomized placebo controlled trials is a limitation that we obviously included in the discussion section.

Some statement about selection of patients is important. Patients with allergy are attracted/referred to doctors with special interests in allergy which tends to increase the apparent prevalence.

The field of interest of a research group cannot condition objective selection/exclusion criteria, as well as the underlying pathologies (endometriosis, IBS-like symptomatology) of a study population.

With all the obvious limitations, a high prevalence obtained can also be the consequence of a probably correct selection and/or a close correlation between the factors investigated.

The idea that Ni acts as a metalloestrogen is interesting but can you link Ni concentration in tissues to estrogenic effects?

The suggestion is truly sensational and, as you suggest and as mentioned in the text, deserves further investigations given the promising results obtained: supporting data already exist in the literature. Not only Ni levels in tissues, but also circulating Ni or Ni contained in the intestinal lumen, could be linked to estrogenic effects. If the theory is confirmed, multidisciplinary studies regarding the effects of metalloestrogens (especially Ni) could be conducted involving endocrinology, gynecology, gastroenterology and even oncology and the alimentary field.

Round 2

Reviewer 2 Report

Thank you for your responses which have addressed my concerns as far as is possible within the limitations of the study design

Author Response

Thank you for your comments, which have surely improved our work.

All changes and modifications have been done, as requested.